# Case Distribution, Sources, and Breeds of Dogs Presenting to a Veterinary Behavior Clinic in the United States from 1997 to 2017

**DOI:** 10.3390/ani12050576

**Published:** 2022-02-25

**Authors:** Katherine H. Anderson, Yufan Yao, Pamela J. Perry, Julia D. Albright, Katherine A. Houpt

**Affiliations:** 1Department of Clinical Sciences, School of Veterinary Medicine, Cornell University, Ithaca, NY 14853, USA; pjp22@cornell.edu (P.J.P.); kah3@cornell.edu (K.A.H.); 2Department of Animal Science, College of Agriculture and Life Sciences, Cornell University, Ithaca, NY 14853, USA; yy675@cornell.edu; 3Department of Small Animal Clinical Sciences, College of Veterinary Medicine, University of Tennessee, 2407 River Drive, Knoxville, TN 37996, USA; jalbrig1@utk.edu

**Keywords:** veterinary behavior, canine, behavioral problems, aggression, anxiety

## Abstract

**Simple Summary:**

This study looked at the reasons why people sought help for their dog’s behavior, where those dogs were acquired by the owner, and which breeds of dogs were seen most often. All dogs in this study presented to a veterinary college specialty hospital in the United States. For dogs presenting to the behavioral medicine service, the types of behavior problems and sources of dogs from 1923 cases over twenty years (1997 to 2017) were evaluated. The breeds of dogs presenting for behavior (822) compared to those presenting to the rest of the hospital for non-behavioral reasons (51,052) were compared over ten years (2007 to 2016). The most common behavioral issue was aggression (72.2% of cases). Dogs obtained from breeders decreased and dogs adopted from shelters increased over twenty years. Dogs from the herding and terrier groups presented more often to the behavioral medicine service compared to the rest of the hospital. While these findings are specific to this population, these results could be helpful in making choices about the allocation of resources for the prevention of behavioral problems, such as genetics research or educational support to shelters. Finally, we found that the variations in terminology and available data made comparisons of behavior problems challenging.

**Abstract:**

The purpose of this retrospective case study was to evaluate trends over time in case distribution, sources, and breeds of dogs presenting to the behavioral medicine service at a veterinary college referral hospital in the United States. For case distribution and sources, the available records from the behavior service (*n* = 1923) from 1997 to 2017 were evaluated. Breeds of dogs presenting to all services (*n* = 51,052) were compared to behavior cases (*n* = 822) from 2007 to 2016. Over twenty years, 72.2% of dogs presented for aggression, 20.1% for anxieties/fears/phobias, and 7.4% for miscellaneous behavioral problems. Dogs acquired from breeders decreased and dogs from shelters, rescues, or adopted as a stray increased over twenty years (*p* < 0.0001). The Herding (*p* = 0.0124) and Terrier (*p* < 0.0001) groups were overrepresented for behavior problems as compared to all other services over ten years. Variations in terminology and diagnostic approach made comparisons with earlier studies difficult, which underscores a need for a more consistent methodology in veterinary behavioral medicine. Understanding trends in sources of dogs could direct resources aimed at guiding owners when acquiring a pet dog and preventing behavioral problems. Findings related to breeds could help guide research focused on the genetic contributions to behavior.

## 1. Introduction

Periodic evaluation of demographic data can be valuable in confirming or refuting clinical impressions as well as achieving better health outcomes in veterinary behavioral medicine [1,2,3]. A current understanding of why owners are seeking behavioral treatment can help guide general practitioners and other specialists in screening and referring cases [4,5,6,7,8], decrease stigma for owners seeking help [9], and focus the clinical training of veterinary students. This is particularly important since education in behavioral medicine is currently limited [10,11]. Understanding trends in how owners acquire pet dogs could direct resources aimed at improving selection criteria and implementing strategies to prevent the development of behavioral problems. Findings related to breeds of dogs presenting more often for behavioral problems could help guide research focused on the genetic contributions to behavior.

Behavioral problems have consistently been shown to diminish the human–animal bond, compromise the welfare of pet dogs, and increase the risk of relinquishment and euthanasia [12,13,14]. However, the classification and accurate description of canine behavioral problems presents significant challenges. Most often, the objectionable behavior is determined by the owner [15,16] and there is currently no standard diagnostic criteria or terminology in veterinary behavioral medicine, making comparisons between cases and populations difficult [17,18]. Additionally, behavior is complex and risk factors for behavioral problems are multi-factorial, including genetics, sex, neuter status, early life experiences, nutrition, acquired learning, training methods, environment, and neurochemistry, among many other features [19,20]. Overemphasis on a single demographic should, therefore, be avoided when determining risk factors [21].

This study aimed to evaluate trends in the case distribution and sources of dogs presenting to the behavioral medicine service at a veterinary college referral hospital from 1997 to 2017 as well as breeds of dogs presenting to the behavioral medicine service compared to the rest of the hospital from 2007 to 2016. Other demographic variables were also evaluated. This project is a follow-up, in part, to Bamberger and Houpt’s 2006 paper on signalment factors, comorbidity, and trends in behavior diagnoses in dogs from 1991 to 2001 [22] with an additional focus on sources of dogs and breed comparisons.

## 2. Materials and Methods

### 2.1. Data Analysis

The available records (*n* = 1923) of dogs that presented to the Animal Behavior Clinic (ABC) at Cornell University Hospital for Animals (CUHA) in Ithaca, NY, USA from 1997 to 2017 were evaluated. The principal behavior problem was identified in 1800 cases. Of those, all 1800 included breed and 1790 included both a behavioral problem and sex (designated as male neutered, male intact, female spayed, and female intact).

Due to variations in terminology, missing data, and limitations of the software used at that time, the cases were grouped according to the following general categories: (1) aggression, (2) anxiety/fear/phobia, (3) miscellaneous, and (4) normal.

The category of “aggression” included cases recorded as human-directed, fear-based, intraspecific, territorial, and predatory aggression. Dogs from the same household presenting for inter-dog aggression were counted as one case for the diagnostic category but their breed and source were listed separately if known. The category of “anxiety/fear/phobia” included separation anxiety, generalized anxiety, and noise aversion (including thunderstorm phobia). “Miscellaneous” included compulsive disorders, pica, self-mutilation, and various nuisance behaviors such as destructive behavior, house soiling, barking, or any other problem that was not designated specifically as either aggression or an anxiety, fear, or phobia. Finally, “normal” included dogs that the owner believed to have a behavioral problem; however, it was determined to be within the normal range of behaviors for the breed, sex, and age of the patient.

Source information was available for 1457 dogs. Sources, as indicated by the owner, included breeder, shelter, rescue, stray, private party, relative/friend, or other. The source categories of “shelter”, “rescue”, and “stray” were combined since the distinction was not always clear based on owner responses. Private-party sources included dogs acquired from farms, advertisements posted in stores, online or in the newspaper, as well as individuals not designated as a friend, relative, breeder, shelter, rescue, or stray.

The available records for breeds presenting to all services in the hospital from 2007 to 2016 (*n* = 51,052) were compared to behavior cases (*n* = 822) for the same period. A total of 217 different breeds were represented. The presented breeds were then categorized into eleven breed groups according to the American Kennel Club (AKC) breed list [23].

The classification of Staffordshire and Pit Bull-type breeds was problematic based on variations in the available data and misidentification of these breeds in general. Entries included American Pit Bull Terrier, Staffordshire Terrier, American Staffordshire Terrier, Staffordshire Bull Terrier, and in some cases, both Staffordshire and Pit Bull Terrier. Therefore, any variation was combined unless they were designated as a mixed breed, for example, “Staffordshire Terrier Mix”. This may have risked an over-representation of Pit Bull-type breeds; however, it is also possible that these breeds are under-represented in general due to breed stigma or breed-specific legislation [24]. For example, an owner may list their dog as a Boxer or Terrier mix to avoid identification as a Pit Bull-type breed. Additionally, visual identification of Pit Bull-type dogs has been shown to be unreliable [25,26,27]. These results should, therefore, be interpreted with caution [28].

### 2.2. Statistical Analysis

Descriptive and univariate statistics were summarized for all key variables. All statistical analyses were completed in JMP Pro (version 15.0.0). Contingency analysis was performed for source and case distribution over time, as well as sex and breed by case distribution.

To examine variation in both sources of dogs and diagnostic categories over time, a chi-square test of goodness-of-fit was performed using annual percentages. Sex and breed were cross-tabulated with the diagnostic categories of aggression, anxiety/fear/phobia, and miscellaneous. Dogs designated as normal were excluded due to small sample size (*n* = 5). Specific breeds compared to diagnostic categories were German Shepherd and American Staffordshire/Pit Bull Breeds, since they presented more often to the behavioral medicine service than all other services in the hospital. A Pearson chi-square test was used for comparisons of sex and breed, with *p* < 0.05 considered significant.

Lastly, breeds and breed groups presenting to the behavioral medicine service were compared to dogs presenting to all other services in the hospital over ten years using a two-tailed Fisher exact test, with *p* < 0.05 considered statistically significant.

## 3. Results

### 3.1. Case Distribution

For dogs presenting to the behavioral medicine service over twenty years, the majority of dogs presented for aggression, followed by anxieties, fears, or phobias, and then miscellaneous behavioral problems. Five cases were designated as “normal” (Table 1). There was no significant variation in case distribution over time (aggression, *p* = 0.3117; anxiety/fears/phobias, *p* = 0.9450; and miscellaneous, *p* = 0.0817).

Separation anxiety was the most common form of anxiety, representing almost half of all anxiety/fear/phobia cases, followed by generalized anxiety. Noise aversion accounted for the fewest number of cases. Unfortunately, secondary problems were not available for evaluation. It was, therefore, not possible to determine if cases presenting for separation anxiety were overrepresented for noise phobia, a common comorbidity [29].

Evaluation of sex and breed by case distribution was fairly unremarkable. Male neutered (*p* = 0.3632), female spayed (*p* = 0.3289), female intact (*p* = 0.0576), German Shepherds (*p* = 0.1966) and American Staffordshire/Pit Bull Breeds (*p* = 0.4352) were not significantly overrepresented for any diagnostic category. However, the relationship between male intact and diagnostic category was significant for aggression (*p* = 0.0026). Further comparisons were challenging due to variations in diagnostic criteria.

### 3.2. Sources of Dogs

The distribution of the source of dogs presenting to the behavioral medicine service changed over time (*p* < 0.0001) with dogs from breeders decreasing and rescue dogs (shelter, rescue, or stray) increasing over twenty years (Figure 1). No significant change was found in dogs acquired from friends/relatives (*p* = 0.9714), private parties (*p* = 0.8455), pet stores (*p* = 0.3718), or other sources (*p* = 0.1773).

### 3.3. Breeds

Findings for breeds and breed groups are summarized in Table 2. Over ten years, mixed-breed dogs presented significantly more often to the behavioral medicine service compared to dogs presenting to all other services in the hospital (*p* < 0.0001). German Shepherds (*p* = 0.0066) and American Staffordshire/Pit Bull-type breeds (*p* < 0.0001) presented more often for behavioral evaluation than for all other services, whereas Golden Retrievers (*p* = 0.0243), Labrador Retrievers (*p* < 0.0001), Shih Tzus (*p* = 0.0296), Beagles (*p* = 0.0040), and Chihuahuas (*p* = 0.0009) presented significantly more often to all other services.

For breed groups, the Herding (*p* = 0.0124) and Terrier (*p* < 0.0001) groups were over-represented in the behavioral medicine service as compared to all other services while Toy (*p* < 0.0001) and Sporting (*p* < 0.0001) groups were under-represented in the behavioral medicine service.

## 4. Discussion

From 1997 to 2017, approximately 70% of dogs referred to the behavioral medicine service presented for aggression and 20% for anxiety, which is similar to the overall case distribution from Bamberger and Houpt’s study from 2006 [22]. Aggression is often cited as the most common presenting problem to veterinary behaviorists [30,31,32,33,34,35]. However, there is likely an inherent bias when looking at dogs presenting for referral based on the geographic location of the hospital, the specific risk the animal poses due to its size or the severity of the behavior problem, as well as the motivation and resources of the owner. For example, owners may be more motivated to seek the help of a specialist for aggression over other behavioral problems, and for larger dogs specifically, due to safety concerns [31,36,37]. Additionally, barriers to seeking treatment might include financial constraints and the owner’s motivation to attend the consultation and implement the recommendations [38].

Studies looking at owner-reported information regarding problem behaviors often vary from those referred to veterinary behaviorists. For example, nuisance behaviors such as barking or digging are often cited more often than aggression [34,39,40]. This may indicate a lower prevalence of aggression in pet dogs in general or it is also possible that owners may be reluctant to describe their dog as aggressive. Dinwoodie et al. reported aggression as the second most common behavioral problem, after fear and anxiety, based on owner surveys [41]. However, since fear and anxiety are common underlying emotions in aggression, it is often difficult to make the distinction between dogs that are aggressive versus anxious (or both) based solely on owner observations.

Furthermore, aggression as a behavioral strategy alone does not provide an explanation for the behavior and therefore cannot be considered a diagnosis. Behavioral problems are often subjective, based on owner preference as opposed to an underlying abnormality. Lack of consensus when approaching a diagnosis is based on many factors, including the number of possible explanations for the behavior, differences in terminology, and variability in diagnostic criteria. Experts do not always agree on the terminology used to classify behavioral problems [42], which makes comparisons across groups difficult. Often, the association between a description of a behavior and the underlying motivation or diagnosis is not straightforward [43,44]. However, whether problematic behaviors are normal but unwanted or due to an underlying pathology, their presence can contribute to decreased welfare for the dog and a weakening of the human–animal bond [12,45,46,47,48,49]. Despite differences in their approach, most experts agree that aggression poses a public safety risk and increases risk of relinquishment and euthanasia [50]. A systematic approach to collecting and evaluating statistics related to canine behavioral problems would allow comparisons between groups, even if a specific diagnosis was not made [51,52].

Looking at the results for sources of dogs referred for behavior problems over twenty years, dogs acquired from rescues and shelters became more common than dogs acquired from breeders. Unfortunately, source information is not available for all dogs presenting to the hospital, so we do not know if this reflects a shift in the preferences of the population at large or if dogs acquired from shelters or rescues are over-represented for behavioral problems. A 2015 survey by Bir et al. at Purdue University cites adoption as the most common means of acquiring a dog [53], while recent data reported by the ASPCA found that slightly more dogs were acquired from breeders [54]. This highlights the need for continued collection of this information in order to know where to focus education and resources on the placement of dogs in compatible homes and reduce the risk of future relinquishment [55,56,57].

In terms of findings for breeds, similarities between this study and Bamberger [22] included mixed breeds and German Shepherds presenting more often for behavioral problems and Labrador Retrievers and Golden Retrievers presenting less often when compared to all other services. In contrast, Duffy et al. found Chihuahuas and Dachshunds ranked higher for aggression based on owner-reported information [58], whereas Guy et al. reported Labrador Retrievers presenting more often for aggression to a general veterinary practice [59]. The differences in our results and owner-reported data could, in part, be related to correlations between the size of the dog and the potential risk they pose. For example, an owner might be more motivated to seek treatment for a larger dog (German Shepherd) over a smaller dog (Chihuahua). Where size is not a factor (i.e., Pit Bull Terrier versus Golden Retriever), one could theorize that genetic predisposition might play a role in the incidence of behavioral problems. However, while there is certainly a genetic contribution to behavior [60] and specific breeds might be over-represented for behavioral phenotypes [61,62], it is important to avoid placing too much emphasis on breed as a sole risk factor [63,64].

## 5. Conclusions

The case distribution in this dataset is consistent with several previous studies looking at referrals to veterinary behaviorists, with aggression as the most common problem, followed by anxiety, most often separation anxiety. In this population, there was a significant increase in dogs adopted from shelters and rescues or acquired as a stray over twenty years while dogs obtained from breeders decreased. Herding and terrier breeds presented more often to the behavioral medicine service over ten years when compared to the rest of the hospital, although this information is also potentially misleading due to confounding factors such as the size of the dogs (more risk of injury) and popular breeds in this geographic area.

How we collect, use, and share information can have a direct impact on the health and welfare of our patients. The challenges in accurately characterizing behavioral problems underscore a need to establish a consistent and reliable classification scheme, even though it is often difficult to put behavior into simple, discrete categories. With a consistent approach, we could ultimately improve our accuracy at risk assessment and prognosis, be better able to make direct comparisons between individuals and groups, and more clearly communicate with other veterinarians, professionals, and the general public.

## Figures and Tables

**Figure 1 animals-12-00576-f001:**
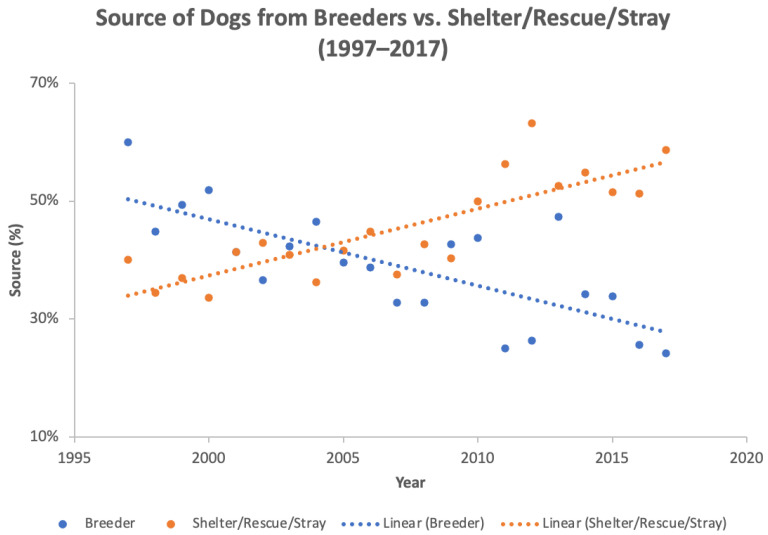
Dogs acquired from breeders versus shelter/rescue/stray for cases presenting to the behavioral medicine service from 1997 to 2017. Dogs from breeders significantly decreased (*p* < 0.0001) while dogs obtained from shelters, rescues, or acquired as a stray significantly increased (*p* < 0.0001) over twenty years.

**Table 1 animals-12-00576-t001:** Behavioral case distribution by category presenting to the Animal Behavior Clinic (ABC) at Cornell University Hospital for Animals (CUHA) from 1997 to 2017.

Diagnostic Category	No. of Dogs (%)	Diagnostic Category	No. of Dogs (%)
Total Aggression	1300 (72.2)	Total Anxiety/Fear/Phobia	362 (20.1)
Human-directed	1037 (79.7)	Separation Anxiety	178 (49.2)
Inter-dog	247 (19.0)	Generalized Anxiety	147 (40.6)
Predatory	9 (0.7)	Noise Aversion	37 (10.2)
Territorial	7 (0.5)	Total Miscellaneous	133 (7.4)
		Total Normal	5 (0.3)

**Table 2 animals-12-00576-t002:** Dog breeds and breed groups presenting to the Animal Behavior Clinic (ABC) compared to all other services at Cornell University Hospital for Animals (CUHA) from 2007 to 2016.

Breed or Breed Group	No. of Dogs (%)		*p* Value
	ABC	CUHA	
Mixed Breed	315 (38.3)	13,053 (25.6)	<0.0001 *
Labrador Retriever	31 (3.8)	5004 (9.8)	<0.0001 *
Golden Retriever	25 (3.0)	2408 (4.7)	0.0243 *
German Shepherd	43 (5.2)	1736 (3.4)	0.0066 *
Dachshund	17 (2.0)	1200 (2.35)	0.7267
Boxer	21 (2.6)	1186 (2.3)	0.6399
Shih Tzu	9 (1.1)	1129 (2.2)	0.0296 *
Beagle	5 (0.6)	972 (1.9)	0.0040 *
American Staffordshire/Pit Bull Breeds	36 (4.4)	1059 (2.1)	<0.0001 *
Chihuahua	3 (0.4)	874 (1.7)	0.0009 *
Bulldog	16 (2.0)	824 (1.6)	0.4041
Herding	92 (11.2)	4414 (8.7)	0.0124 *
Toy	36 (4.4)	6196 (12.1)	<0.0001 *
Sporting	102 (12.4)	10,131 (19.8)	<0.0001
Non-Sporting	61 (7.4)	3778 (7.4)	0.9465
Working	81 (9.9)	5715 (11.2)	0.2262
Terrier	87 (10.6)	3404 (6.7)	<0.0001 *
Hound	44 (5.4)	3397 (6.7)	0.1571

* Significance of *p* < 0.05.

## Data Availability

Please contact the first author for supporting data.

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
