# Peer review of "Case Distribution, Sources, and Breeds of Dogs Presenting to a Veterinary Behavior Clinic in the United States from 1997 to 2017"

_animals, 2022, doi:10.3390/ani12050576_

Round 1

Reviewer 1 Report

Case distribution, sources, and breeds of dogs presenting to a veterinary behavior clinic in the United States from 1997-2017

The manuscript is beautifully written, good job! It is also important data for veterinarians and behaviorists. There are a few suggestions I have made below that I think would make more use of your valuable data. Great job overall.

L111-113: On Statistical analysis, why didn’t you compare demographics data to types of behaviors dogs presented? Moreover, isn’t breed and breed group also part of demographics? If race and ethnicity are part of demographics data for humans, I would suppose the same applies to dogs.

Figure 1: The correlation over the years is not described under M&M. I would suggest you have a subsection for “Statistical Analysis”.

Figure 1: This figure is very helpful and nicely done. Could you add a similar figure(s) with the amount of cases of anxiety and aggression over the years? We might see an increase in anxiety over the years- this information would be very useful.

Table 2: Please add the meaning of your abbreviations either on the title or as a footnote. The Tables need to be self-explanatory. Also, I think it is obvious, but you compared the percentages of each group, not the actual numbers, right? That needs to be added to stats description.

L166-167: this hyphen does not work well- could you maybe split this phrase in two?

Discussion- I would like to see some discussion on how behavioral problems changed over the years (this related to previous comment on Figure 1).

Author Response

Thank you for taking the time to review our paper. Please see the attachment for our responses. I also updated the manuscript to reflect these changes.

Reviewer 2 Report

The study is original and deserves publishing. However, the manuscript would benefit from a clearer description of the methods (especially, statistical analysis), as well as further discussion of the results. I added further/detailed comments and recommendations in the file attached.

Author Response

Thank you for taking the time to review our paper. Please see the attachment for our responses.
